# Construction of an Electrical Conductor, Strain Sensor, Electrical Connection and Cycle Switch Using Conductive Graphite Cotton Fabrics

**DOI:** 10.3390/polym14214767

**Published:** 2022-11-07

**Authors:** Fahad Alhashmi Alamer, Asal Aldeih, Omar Alsalmi, Khalid Althagafy, Mawaheb Al-Dossari

**Affiliations:** 1Department of Physics, Faculty of Science, Umm AL-Qura University, Al Taif Road, Makkah 24382, Saudi Arabia; 2Department of Physics, Faculty of Applied Science, King Khalid University, Abha 62529, Saudi Arabia

**Keywords:** graphite, cotton fabrics, sheet resistance, strain sensor, cycle switch

## Abstract

Researchers in science and industry are increasingly interested in conductive textiles. In this article, we have successfully prepared conductive textiles by applying a graphite dispersion to cotton fabric using a simple brush-coating-drying method and the solvents of dimethyl sulfoxide, dimethyl formamide, and a solvent mixture of both. The sheet resistance of the resulting cotton fabrics could be influenced by the type of polar solvent used to prepare the graphite dispersion and the concentration of graphite. In addition, the graphite cotton fabrics showed semiconductive behavior upon studying the resistance at different temperatures. A flexible strain sensor was fabricated using these graphite cotton fabrics for human motion detection. Most importantly, the resulting strain sensor functions even after 100 bending cycles, indicating its excellent reproducibility. In addition, our results have also shown that these graphite cotton fabrics can be used as electrical interconnects in electrical circuits without any visible degradation of the conductive cotton. Finally, a cotton electrical cycle switch was made using the graphite cotton fabrics and worked in the on and off state.

## 1. Introduction

In recent years, researchers have become interested in conductive materials because they can be used in flexible and wearable devices that make life safer, healthier and more convenient [1,2,3]. Conductive fabrics are ideal fabric candidates for the design of electronic textiles (e-textiles) [4,5,6,7,8,9] and form the basis of future technologies such as carbon-based conductive fabrics [10,11,12,13]. This study investigates the use of graphite to produce conductive cotton fabrics. Graphite has good environmental stability and excellent electrical conductivity and is commercially available. It can transport electrons rapidly [14], making it a potential component candidate for a variety of applications, such as supercapacitors [15], solar cells [16,17], batteries [18] and portable electric heaters [19,20,21]. These products must be lightweight, flexible and durable to be functionally utilized [22,23]. In addition, e-textiles are important for the fabrication of strain and bending sensors [24,25,26,27,28], which have shown promise for biomedical monitoring in the fields of health and sport, such as tracking body movement, monitoring levels of activity and recording improvements in joint movements [29].

Among the commercially available fabrics, cotton fabrics are widely used in the field of e-textiles because they are comfortable, breathable, highly absorbent, flexible and lightweight. Other advantages of cotton fabric include its low material and processing costs. Importantly, cotton fabrics have high porosity, and their chemical structure includes surface functional groups such as hydroxyl and carbonyl groups, which improves the bonding and adhesion of conductive materials to the fabric [30].

Conductive cotton fabrics are designed and manufactured in one of two ways. The first method uses conductive polymers, which have the advantages of flexibility, durability and relatively high conductivity [31,32,33,34]. However, the large-scale fabrication and poor solubility of conductive polymers in solvents are drawbacks of this approach. The second method incorporates conductive materials into fabrics [35,36,37]. Materials comprising carbon atoms are ideal candidates for this purpose and provide good conductivity, high sensitivity, and preservation of the mechanical properties of the fabric. Typical materials include carbon black [38,39], carbon nanotubes (CNTs) [40,41,42], graphene and graphite [43,44] and are applied by coating methods such as electrodeposition [45,46,47], dip-coating [48,49], drop casting [43] and chemical vapor deposition (CVD) [50,51,52].

Woltornist et al. [44] prepared conductive fabrics using an interface trapping technique with natural flake graphite. They found that the sheet resistance of the fabrics changed from 77.9 M Ω/□ to 3.6 KΩ/□ when the graphite concentration increased from 2.5 wt% to 7.4 wt%. They also observed that electrical resistance as a function of temperature showed an insulator–metal transition and that the fabrics had the lowest resistance near room temperature. Chatterjee et al. [53] produced conductive fabrics by first immersing the materials in a graphene oxide (GO) solution and then subjecting the fabrics to a reduction process. They used woven and knitted fabrics and found that the concentration of graphene oxide solution and the number of immersion cycles affected the sheet resistance of the conductive fabrics. After 15 immersion cycles, they found that woven and knitted fabrics with a concentration of 2.25% GO had sheet resistances of 0.26 MΩ/□ and 0.19 MΩ/□, respectively, and attributed the differences in sheet resistance to the fact that the fabrics contained different amounts of graphene.

Kim et al. [54] prepared a conductive textile using a composite of graphene/waterborne polyurethane and the dip-coating method. They found that the surface resistivity of the textile depended on the number of dip-coating cycles and decreased from $4.0\times 10^{9}\Omega /□$ to $1.5\times 10^{3}\Omega /□$ when the number of coating cycles increased from one to five and attributed this behavior to an increased amount of graphene retained by the textile. In addition, they reported that the electrical capacitance of the textiles increased from 3.4 pF to 19.8 pF as the number of coating cycles increased from one to five. In another study [55], a composite of polyacrylonitrile/graphite was used to coat textile fabrics using the doctor blade technique. The results showed that the resistance depended on the thickness of the coating layer and changed from 4 kΩ to 10 kΩ for the thicker coating and from 20 kΩ to 50 kΩ for the thinner coating. Mizerska et al. [56] coated cotton fabric with an organosilicon solution (sol) containing dispersed graphene oxide (GO) by the sol-gel method followed by thermal treatment to induce the reduction of GO. The main purpose of this process was to improve the electrical conductivity and hydrophobicity of the cotton fabric. They found that the sheet resistance of the sample coated with diluted sol and graphene oxide decreased from 94 MΩ/□ to 6.70 MΩ/□ after 24 h of thermal treatment, when the content of reduced graphene oxide increased from 1.7 wt% to 3.2 wt%. In addition, they reported that the sheet resistance of the sample coated with 0.5 wt% graphene oxide alone had a lower value of 0.74 MΩ/□. They attributed this to the partial isolation of the network-forming particles in the organosilicon coating caused by the absence of the polymer. They also observed that the sample coated with dilute sol and graphene oxide exhibited greater hydrophobicity than the sample coated with graphene oxide alone.

In the present study, a simple “brush-coating-drying method” was developed to prepare conductive cotton fabric with graphite. The polar solvents DMSO, DMF and a mixture of both were used to disperse the graphite. The graphite dispersions were applied to three cotton fabric samples with a brush and then dried, and the process was repeated until a saturated coating was achieved. The effects of the polar solvents, graphite concentration and temperature on the sheet resistance of the cotton fabric samples were investigated. The specific results were as follows: First, the saturation concentration of 86.12 wt% graphite/DMF-coated cotton fabrics resulted in a sheet resistance of 7.975 kΩ/□. Second, the saturation concentration of 66.85 wt% graphite/DMF-coated cotton fabrics yielded a sheet resistance of 2.676 kΩ/□. Finally, the conductive cotton fabrics prepared with graphite and a solvent mixture gave the best performance with a low sheet resistance of 1.197 kΩ/□ and a lower graphite amount of 58.70 wt%. The feasibility of graphite-coated cotton fabrics as strain sensors was also investigated. This investigation showed that electrical conductivity was maintained for more than 100 bending cycles. In addition, the graphite-coated cotton fabrics were used as a conductive connection in an electrical circuit without any visible degradation of the conductive cotton. Finally, the graphite-coated cotton fabrics were used to fabricate an electrical cotton cycle switch that mimicked the real cycle switch in electrical circuits and worked in the on and off states. Table 1 shows that the sheet resistance values of our work are lower than those reported in the literature for different fabrics infused with carbon-based materials, and how many bending cycles the fabrics used as strain sensors undergo.

## 2. Experimental

### 2.1. Materials and Characterization

In this study, graphite powder, dimethylsulfoxide (DMSO) and dimethylformamide (DMF) were purchased from Sigma Aldrich and used without purification. A pure cotton fabric was obtained from a local store. To calculate the sheet resistance of the conductive graphite cotton fabrics, the electrical resistance R was first calculated from an I-V curve using the four-line probe technique and constructed according to the literature [60]. The sheet resistance *R_s_*, in units of Ω/□, was then calculated using the relation $R_{S}=R\left(\frac{w}{L}\right)$, where w represents the sample width (2.5 cm) and L represents the distance between the probes (0.35 cm). Using a Keithley 2400 current source meter and an HP 34,401 A multimeter, we generated the required amount of current and measured the resulting potential difference. Scanning electron microscopy (SEM) was carried out using a Thermo Scientific Scios 2 scanning electron microscope. The FTIR spectra were recorded using an FTIR instrument, an IR Spirit spectrometer with a QATR-S accessory, in the range 400–4000 cm^−1^, which allowed fast and easy analysis of the functional groups in the graphite powder, untreated cotton, and conductive graphite cotton. For this instrumental analysis, the samples were not mixed with potassium bromide. Thermogravimetric analysis (TGA) was performed on the graphite powder and the treated and untreated samples using a TG/DTG-60/60H instrument in the temperature range from 35 °C to 1000 °C and at a rate of 50 °C/min under nitrogen purge. A Bruker D8 ADVANCE XRD (X-ray diffractometer) in 2-theta geometry was used to record the XRD patterns of all materials. Intensities were recorded over a 2-theta range from 5° to 80°. A Cu-Kα (l 1/4 1.540598) radiation source was used for the measurements.

### 2.2. Preparation of Solutions

Three different experimental solutions were developed for the production of graphite cotton fabrics. Solution I: One gram of graphite was mixed with 4 mL of deionized water. The mixture was then sonicated at room temperature for 10 min after the addition of 4 mL DMF. Solution II: Four ml of deionized water was mixed with 1 g of graphite, followed by 4 mL of DMSO. The solution was then sonicated at room temperature for 10 min. Solution III started with a mixture of 4 mL of deionized water and 1 g of graphite, to which a solvent mixture of DMSO and DMF (1:1) was added. Then, the resulting solution was sonicated at room temperature for 10 min [61,62,63,64].

### 2.3. Preparation of Conductive Graphite Cotton Fabrics

In this work, we have developed an effective, simple method to prepare conductive graphite cotton fabrics using the “brush-coating-drying” technique at room temperature and under ambient air conditions (see Figure 1). First, a cotton fabric sample was dipped in deionized water to improve the absorption of the graphite solution. Then, a brush was dipped into the graphite solution, and the wet cotton fabric was quickly coated with graphite covering the entire surface. Finally, the treated cotton fabric samples were oven dried at 130 °C for 20 min. The graphite concentration in the fabric sample was determined after the drying phase using the formula, $C\left(wt.\%\right)=\frac{C_{2}-C_{1}}{C_{1}}\times 100$ where C_2_ represents the weight of the treated sample and C_1_ represents the original weight of the (untreated) sample. The brush-coating-drying procedure was repeated to increase the amount of graphite in the cotton until it reached the saturation concentration, i.e., when adding more graphite dispersion to the cotton resulted in a thick layer on the surface of cotton. The preparation of the three different conductive cotton samples can be summarized as follows.

Sample I: A brush was soaked in solution I, and then a cotton fabric sample was coated using the soaked brush. The sample was then dried in the oven for 20 min at 130 °C. Then, we repeated the coating process approximately 15 times to reach the saturation concentration. Sample II: A brush was soaked in solution II, and then the cotton fabric sample was coated using the soaked brush. After that, the sample was dried in an oven for 20 min at 130 °C. The saturation concentration was reached by repeating the coating procedure approximately 8 times.

Sample III: A brush was soaked in solution III, and then the cotton fabric sample was coated using the soaked brush. After that, the sample was dried in an oven at 130 °C for 20 min. The saturation concentration was reached by repeating the coating procedure approximately 11 times.

## 3. Results and Discussion

### 3.1. Morphological Observations

SEM was used to investigate the morphological structures of the graphite and the treated and untreated cotton fabrics without any pretreatment, so we consider this characterization to be an environmentally friendly treatment. Figure 2 shows the corresponding SEM images. Figure 2a shows an SEM image of the untreated graphite with dense flakes approximately 5 μm in diameter. Figure 2b is an SEM image of the untreated cotton showing that the untreated cotton is woven and contains fibers and gaps between each fiber. The enlarged image (at 1200×) of the same surface (Figure 2c) shows a relatively smooth, elongated fibril structure with no visible impurities. The SEM images of the treated cotton fabrics are shown in Figure 3 at different graphite concentrations from low to high. The surface of the treated cotton samples featured ripples compared to the smooth and clean surface of the untreated cotton. Figure 3a,d,g show the existence of graphite, in low concentrations, on the fiber surface, creating conductive paths. If the graphite concentration was slightly increased, there was a potential for more graphite to be present on the surface of the fibers and in the gaps between the fibers (see Figure 3b,e,h). At high concentrations, graphite was encapsulated on the surface of the fibers and in the area around each fiber, as shown in Figure 3c,f,i. This indicates that the graphite at high concentrations may have created a larger guiding path for the charge carriers compared to the graphite at low concentrations.

As seen in the SEM images in Figure 4, the graphite particle size in sample I (Figure 4a is larger than the graphite particle sizes in sample II and sample III (Figure 4b,c). This is due to the DMF solvent increasing the adhesion of the graphite particles to each other, which leads to an increase in particle size and explains the reason for the decrease in sheet resistance in sample I, as will be discussed later.

### 3.2. XRD Analysis

The XRD patterns of the graphite powder and the untreated and treated cotton fabrics are displayed in Figure 5 and Figure 6. The lattice spacing d was calculated from Bragg’s law nλ=2dsinθ where n is an integer, λ is the wavelength of X-rays and θ is the diffraction angle. The average grain size ($D_{c})$ of the graphite was calculated from Scherrer’s equation $D_{c}=\frac{0.94\lambda }{\beta cos\theta }$ where β is the line broadening at the half maximum intensity, in radians. The XRD pattern of the graphite powder in flake form (Figure 5A) shows characteristic peaks of (002) at 26.42°, (101) at 44.46° and (004) at 54.51° with high intensity which is hardly observed. The lattice spacings d 002, d 101 and d 004 were determined to be 0.337 nm, 0.203 nm, and 0.168 nm, respectively and the average grain size of the graphite was calculated to be 17.01 nm [65]. Figure 5B shows the XRD pattern of the untreated cotton sample with preferential orientation along the c-axis. The main characteristic peaks were observed at 14.99°, 16.49°, and 22.78°, indicating Miller indices (1–10), (110), and (200), respectively. The moderate peak at 34.5° was a complex of several weak peaks that overlapped, with (004) not being the main factor [66]. The lattice spacings d 1–10, d 110, and d 200 were determined to be 5.91 nm, 5.37 nm, and 3.90 nm, respectively (see Table 2). The average grain size of the untreated cotton fabrics was calculated to be 11.13 nm. The XRD results of the flake graphite powder and untreated cotton are in good agreement with JCPDS file numbers 00-008-0415 and 00-050-2241, respectively (Table 2). The graphite-treated cotton fabric XRD patterns for samples I, II, and III are similar to those of the untreated cotton fabrics, with an additional peak at 26.42 (see Figure 6) due to the attachment of graphite to cotton fibers. However, the low graphite concentration of sample I resulted in XRD patterns that were relatively unaffected by the treatment, in which only the three characteristic peaks of untreated cotton fabrics appear. The lattice spacing and average grain size of the treated cotton samples at different concentrations are shown in Table 2.

### 3.3. FTIR Analysis

FTIR spectra in the range of 4000 to 400 cm^−1^ were used to analyze the chemical structures of the graphite, untreated cotton fabric, and treated cotton fabric, as shown in Figure 7. In the graphite spectrum, the absorption peaks are located at ~2400 cm^−1^ and ~1580 cm^−1^, which correspond to the OH stretching vibrations [67] and the skeletal vibrations of the graphite chain [68] respectively. The characteristic absorption bands for pure cotton are the hydrogen-bonded OH stretching at 3331.03 cm^−1^, the C-H stretching of the ß-glucose unit of cellulose at 2902 cm^−1^, the C=O stretching of a carboxylic acid and an ester at 1729 cm^−1^, the C=O stretching of an acid salt at 1533 cm^−1^, and the C-O stretching at 1036.22 cm^−1^. As shown in Figure 7, some characteristic absorption bands of the graphite-interspersed cotton fabric (samples I to III) are similar to those of the pure cotton and graphite. However, there are remarkable changes in the FTIR spectrum of the graphite-infused cotton fabric. The C-C stretching at 650 cm^−1^ in the FTIR spectrum of the pure cotton disappears in the spectrum of the graphite-infused cotton fabrics, indicating the binding of graphite to the cotton fabric. It is also shown that the intensity of the peaks decreases after the infusion of graphite into the cotton. All these remarkable changes indicate the binding of graphite to the cotton fabrics.

### 3.4. Electrical Study

#### 3.4.1. Sheet Resistance Measurements

The sheet resistance was investigated for samples I, II, and III to test the effect of the graphite concentrations in different solutions on the electrical properties of the cotton fabric, as shown in Figure 8. At first glance, it is obvious that the sheet resistance of the treated cotton fabrics decreases as the graphite concentration increases. For sample I, the sheet resistance drops dramatically by three orders of magnitude from 5.325 MΩ/□ to 7.975 kΩ/□ when the graphite concentration increases from 8.186 wt% to 53.542 wt%, as shown in Figure 8a. Furthermore, raising the graphite concentration beyond this point had no discernible effect on sheet resistance. However, the minimum sheet resistance obtained for sample I was 7.975 kΩ/□ at 86.124 wt% (saturation concentration).

For sample II, the sheet resistance decreases dramatically by two orders of magnitude from 8.917 MΩ/□ to 2.676 kΩ/□ when the graphite concentration increases from 8.541 wt% to 66.854 wt%, as shown in Figure 8b. The saturation concentration of sample II was set at 66.854 wt% graphite, since graphite solutions at concentrations above this value do not penetrate into the cotton fabric. For sample III, the sheet resistance decreases by an order of magnitude from 0.2058 MΩ/□ to 0.0236 MΩ/□ when the graphite concentration increases from 7.135 wt% to 58.702 wt%, as shown in Figure 8c. Then, the sheet resistance decreases with increasing graphite concentration until it reaches a minimum value of 1.197 kΩ/□.

#### 3.4.2. Sheet Resistance Theoretical Analysis

The main purpose of sheet resistance theoretical analysis is to determine the nature of the relationship between $R_{s}$ and C. Therefore, we have plotted the natural logarithm of $R_{s}$ versus C and the natural logarithm of $R_{s}$ versus the natural logarithm of C, as shown in Figure 9 and Figure 10, respectively. Then, we determined the equations of the line of fit and the related R^2^ values.

First, the fitting of the relation between ln $R_{s}$ and *C* (Figure 9a,b) generated an approximately straight line of fit with high R^2^ value, which indicates that $R_{s}$ is exponentially related to *C* according to the following equation:(1)$$R_{s}=Ae^{-\alpha C}$$
where A and α are the fitting parameters in Table 2. In contrast, with regard to high graphite concentrations, Figure 9c shows another good linear fit between ln $R_{s}$ and *C* with a likewise high R^2^ value, indicating, however, that the resistance hardly changes at low graphite concentrations. Second, the fitting of the relation between ln $R_{s}$ and ln *C* (Figure 10a,b) generated a good straight line of fit, which indicates that R is inversely proportional to $C^{\alpha }$ according to the following equation:(2)$$R_{s}=\frac{A}{C^{\alpha }}$$
where A and α are the fitting parameters in Table 3. Figure 10c depicts two distinct behaviors as the concentration increases. The plots of ln *R* and ln *C* show little change in ln *R* with regard to ln *C* for low values of *C*. However, for high values of *C*, a good straight line with high R^2^ values exists, and this behavior is correlated to Equation (2).

#### 3.4.3. Graphite Cotton as an Electrical Conductor

To decide which cotton sample was best suited as an electrical conductor, the efficiency of each sample was evaluated, and the amount of graphite required to produce specific values of sheet resistance was determined. The log $R_{s}$ on the *x*-axis is plotted against the graphite concentration on the *y*-axis in Figure 11. When compared between the samples, it is clear that the conductive cotton treated with a mixture of graphite and DMF (Sample I) uses graphite most efficiently to obtain a given sheet resistance, as it has the lowest sheet resistance at the same graphite concentration. However, there is a small region of low resistance (labeled D in Figure 11) where the conductive cotton treated with a mixture of graphite and DMSO (sample II) is more efficient than the other two samples, achieving low resistances with significantly less graphite.

#### 3.4.4. Temperature-Dependent Sheet Resistance

The temperature-dependent electrical sheet resistance of samples I and III was measured from 25 °C to 153 °C, and that of sample II was measured from 25 °C to 180 °C. Figure 12 depicts the temperature-dependent sheet resistance of samples I (86.12 wt.% graphite), II (66.85 wt.% graphite), and III (84.97 wt.% graphite) at different temperatures. All samples exhibit the same temperature-dependent trend, with the sheet resistances of samples I, II, and III decreasing exponentially with increasing temperature, indicating semiconductive behavior.

### 3.5. Thermal Analysis

The thermal stabilities and degradation profiles of the graphite powder, pure cotton, and treated samples were investigated using TGA thermograms in which the temperature was raised from room temperature to 1000 °C with a constant ramp rate of 10 °C/min^−1^ under nitrogen flow through the sample chamber (see Figure 13). First, a graphite powder sample (~10 mg) was used to determine the onset temperature of decomposition. The TGA curve shows that the weight loss due to combustion starts at approximately 675 °C [33], and the temperature of the onset of intense thermal decomposition is approximately 983.18 °C. This result is consistent with data published in the literature for graphite powder. A TGA experiment was then performed on the untreated and treated cotton to determine how the infusion of graphite affects the thermal stability and degradation profiles of the produced conductive fabrics. The first loss of mass for the untreated and treated cotton samples occurs at approximately 115 °C, which is due to the release of adsorbed water from the cotton samples, as shown in Figure 13. The onset of the melting peak occurs gradually from 300 °C for untreated cotton, 310 °C for sample I, and 290 °C for samples II and III. The major decomposition of the untreated cotton control peak begins after 340 °C and continues until the maximum melting temperature is reached at 410 °C, with a mass loss of 56 wt%. For samples I and II, the major decompositions of the control peaks start after 350 °C and 330 °C, and the corresponding peaks melt at 421 °C and 410 °C with mass losses of 38.5 wt% and 37.5 wt%, respectively. For sample III, the decomposition starts after 330 °C, where two steps of weight loss were observed, the first step of approximately 31.42 wt% at 396 °C, and the other step of 53 wt% starts at approximately 443 °C, corresponding to the melting peak. In contrast, the graphite at 700 °C showed a significant weight loss due to combustion [25]. Table 3 shows more information about the onset temperature, endset temperature, maximum degradation temperature of the processes, and the weight loss for all samples. Table 4 show that graphite powder and pure cotton fabrics are thermally stable in the ranges from 100 °C to 800 °C and 100 °C to 340 °C, respectively. Sample I is thermally stable up to 350 °C, while samples II and III are thermally stable up to 343 °C. Thus, when graphite is introduced into cotton fabrics, the thermal stability of the cotton increases by 3 to 10 °C.

### 3.6. Electrical Applications of Graphite Cotton Fabrics

#### 3.6.1. Wearable Flexible Strain Sensor

A strain sensor was made from a square piece of conductive cotton fabric with an area of 1 in^2^ and 86.12 wt% graphite to monitor human physiological responses in real time, especially small movements such as finger flexion (see Figure 14a). First, the treated cotton was attached to a pristine cotton substrate. Then, two copper bands were attached as electrodes to the two ends of a piece of conductive cotton. Finally, the electrical resistances of the bending sensors were measured at different bending positions with a two-probe method using a digital multimeter. To investigate the electromechanical properties of the sensor, the electrical resistance was measured at different tensile voltages. The electrical resistance of the graphite/cotton sensor decreased from 7.91 kΩ to 4.92 kΩ when the sensor was subjected to tensile stress with different concave radii from 3 cm to 1 cm, as shown in Figure 14b. The decrease in electrical resistance values could be due to the creation of new electrically conductive pathways. To fully demonstrate the performance of the graphite/cotton fabric as a strain sensor, its response to tensile loading was investigated in three moods: relaxing mood, mild bending mood, and strong bending mood, as shown in Figure 15a. The graphite/cotton fabric was glued on a finger, and the two ends were connected with a multimeter. Then, the change in electrical resistance of the graphite/cotton sensor was recorded in the three different moods. When the mode was relaxed, the resistance of the graphite/cotton sensor increased up to 4 kΩ, and then the resistance decreased to approximately 1.7 kΩ when the mode was slightly bent. In addition, the resistance decreased further to approximately 0.8 kΩ with severe bending. As shown in Figure 15b, the graphite/cotton sensor was very sensitive to repeated tensile stresses. It was observed that the resistance changed repeatedly following the finger bending in the three moods even after approximately 100 bending cycles. Therefore, we conclude that the graphite/cotton fabric made in this way has the reproducible property of a strain sensor.

#### 3.6.2. Conductive Interconnection

A sample containing 86.12 wt% graphite/cotton was used as a conductive interconnection in an electrical circuit, as shown in Figure 16, which includes an LED and a power supply, to vividly demonstrate the electrical conductivity of graphite/cotton fabric. It has been shown that the graphite/cotton fabric that closes the circuit can turn on LEDs. Therefore, graphite/cotton fabric should, in theory, be able to be used as a flexible electrode.

#### 3.6.3. Electrical Cycle Switch

Figure 17a shows how the electrical cycle switch was made for this experiment. Untreated cotton was placed between two pieces of graphite/cotton fabric, with a square hole cut above the untreated cotton. Then, two copper strips were connected as electrodes to the two ends of the graphite/cotton sample. Finally, two thin layers of cotton were placed on top and bottom of the device as insulating layers. When the switch was pressed, the two graphite/cotton pieces connected, and the current flowed in the circuit (ON operation of the device). Then, the LED consumed the current from the source, as shown in Figure 17b. When the two graphite/cotton parts were opened by removing the pressure, the LED did not consume current, as shown in Figure 17c.

## 4. Conclusions

In summary, a conductive textile was successfully fabricated by applying a graphite dispersion to cotton fibers using a brush-coating-drying process. The choice of solvent and graphite concentration had a significant effect on the sheet resistance of the resulting conductive cotton fabrics. The minimum sheet resistance was 0.47 kΩ/□ for the cotton fabric prepared with graphite dispersed in DMF. The strain sensor test showed that the graphite cotton fabrics have excellent reproducibility and can withstand bending processes even after 100 bending cycles. The results suggest that the graphite-embedded cotton fabric sensor has significant potential for wearable sensing applications. Graphite cotton fabric has been used as a component in electrical circuits containing LEDs. The graphite cotton fabrics exhibited very stable conductivity, and the LED could be operated at full intensity without damaging the cotton fabrics. In addition, a fabricated graphite–cotton cycle switch was used as a component in an electrical circuit that operates in the on and off states. We anticipate that graphite cotton fabrics generated using this technology will surely be promising candidates for the production of various fabric-based functional electrical devices due to their superior electrical conductivity and mechanical flexibility.

## Figures and Tables

**Figure 1 polymers-14-04767-f001:**
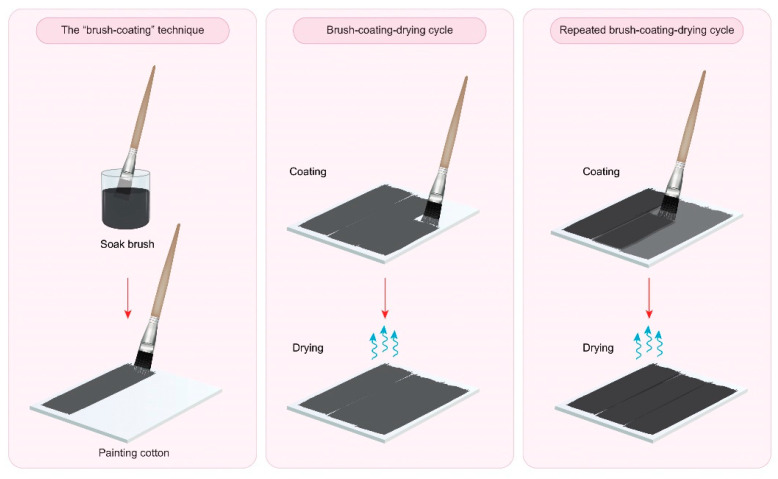
Diagram of “brush-coating-drying” technique.

**Figure 2 polymers-14-04767-f002:**
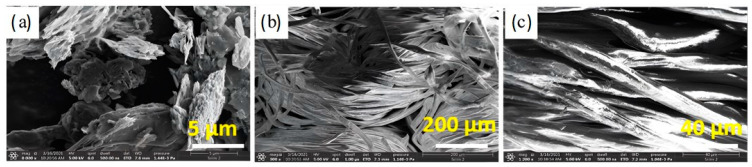
SEM Images of (**a**) graphite powder (**b**,**c**) untreated cotton fabrics at two different magnifications.

**Figure 3 polymers-14-04767-f003:**
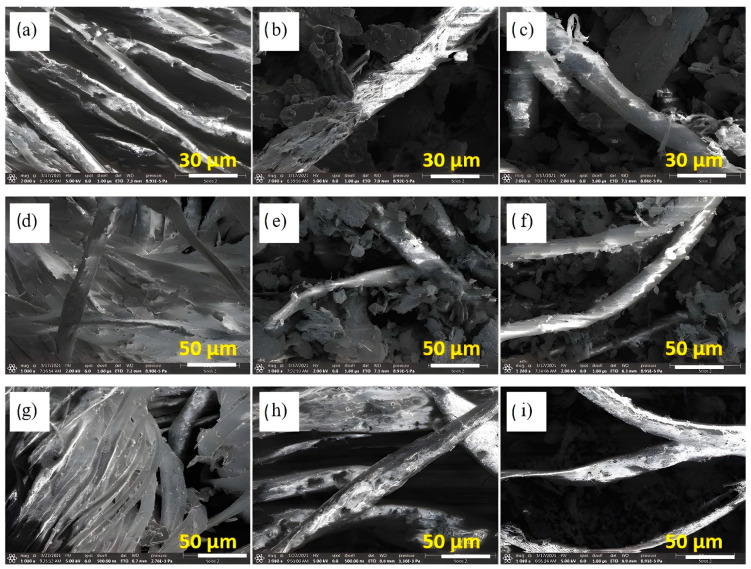
SEM images (**a**,**d**,**g**) at low concentration, the images (**b**,**e**,**h**) at medium concentration and the images (**c**,**f**,**i**) at high concentration for sample I, II, and III respectively.

**Figure 4 polymers-14-04767-f004:**
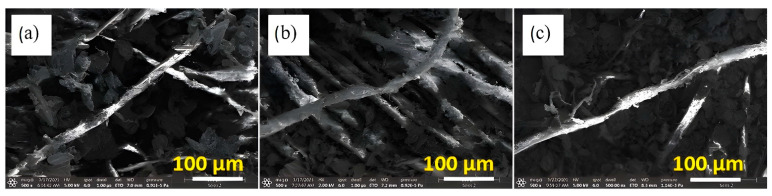
SEM images show the graphite particle in sample I (**a**), sample II (**b**) and sample III (**c**) respectively.

**Figure 5 polymers-14-04767-f005:**
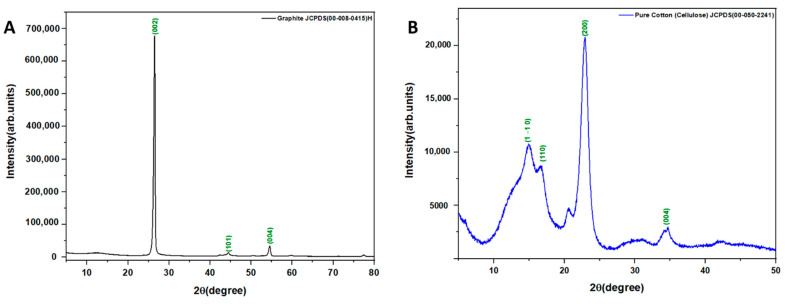
XRD patterns of (**A**) graphite powder (**B**) untreated cotton fabrics.

**Figure 6 polymers-14-04767-f006:**
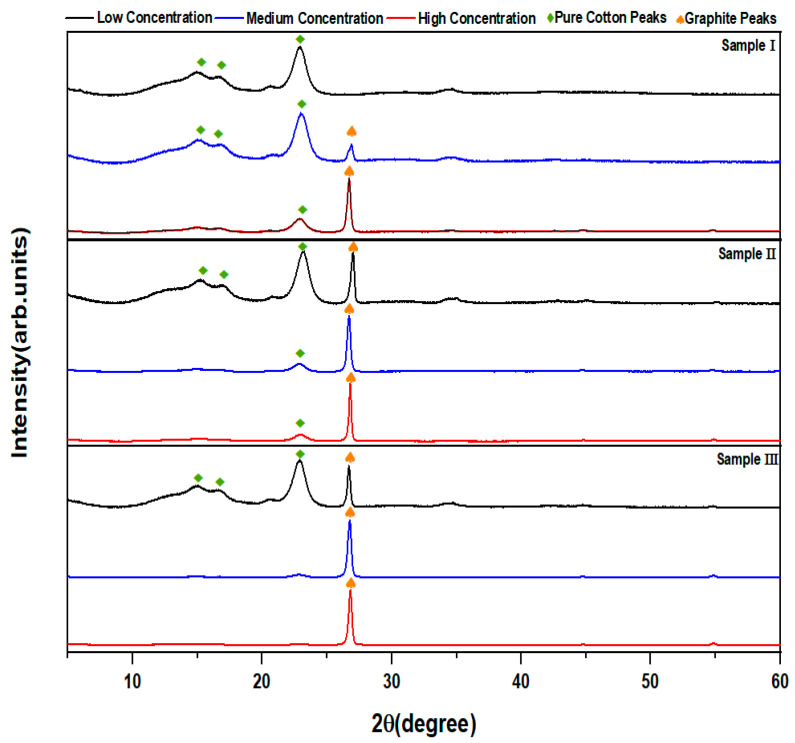
XRD patterns for sample I, II and III at low, medium and high concentrations.

**Figure 7 polymers-14-04767-f007:**
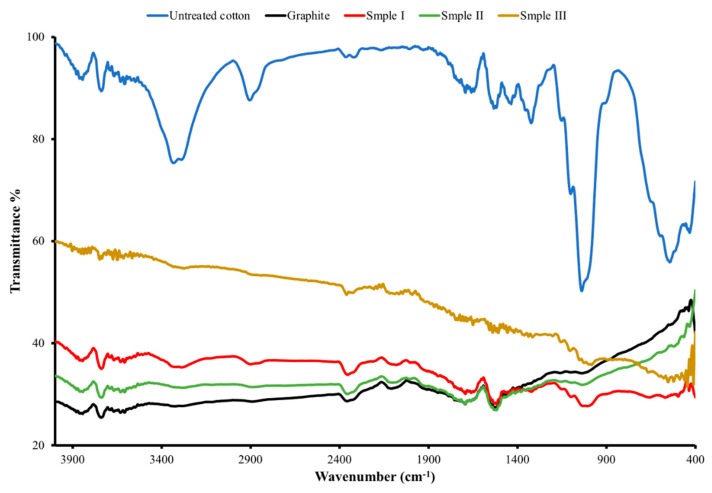
FTIR spectrum of graphite, untreated and treated cotton fabrics (sample I, sample II and sample III, respectively).

**Figure 8 polymers-14-04767-f008:**
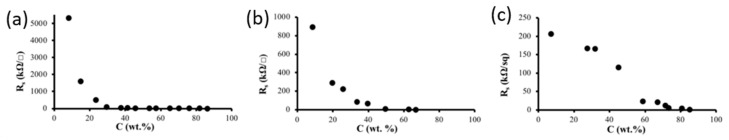
Sheet resistance of sample I (**a**), sample II (**b**) and sample III (**c**) at different graphite concentrations.

**Figure 9 polymers-14-04767-f009:**
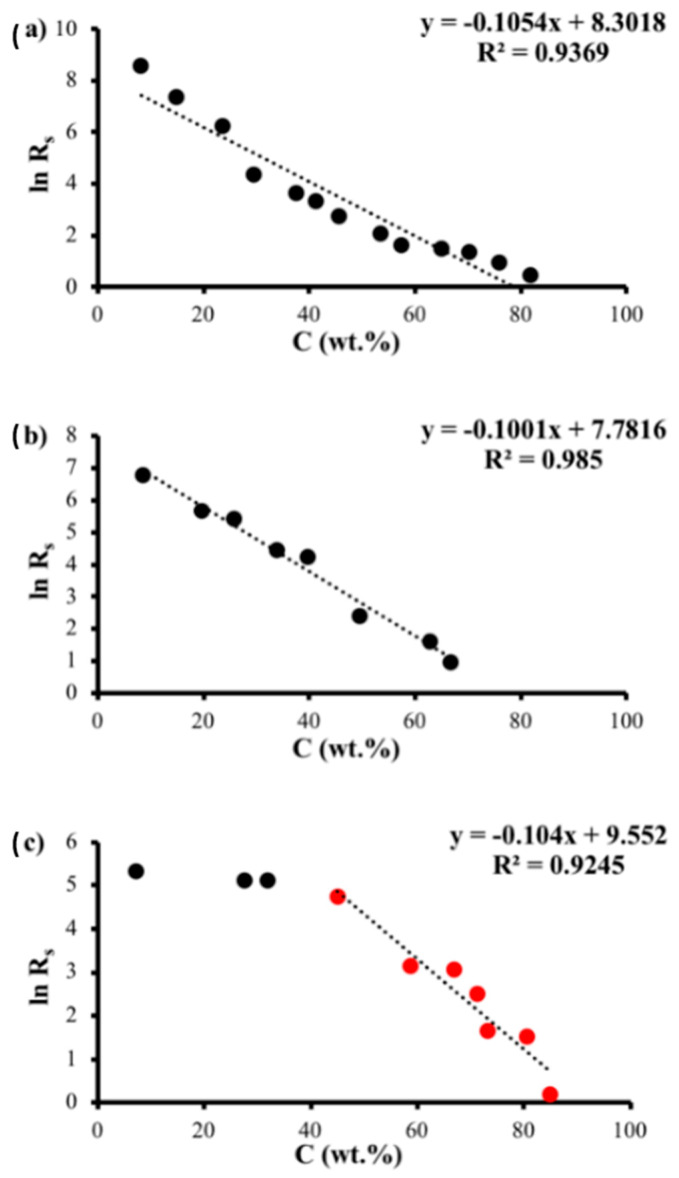
Natural logarithm of sheet resistance of sample I (**a**), sample II (**b**) and sample III (**c**) as a function of graphite concentration.

**Figure 10 polymers-14-04767-f010:**
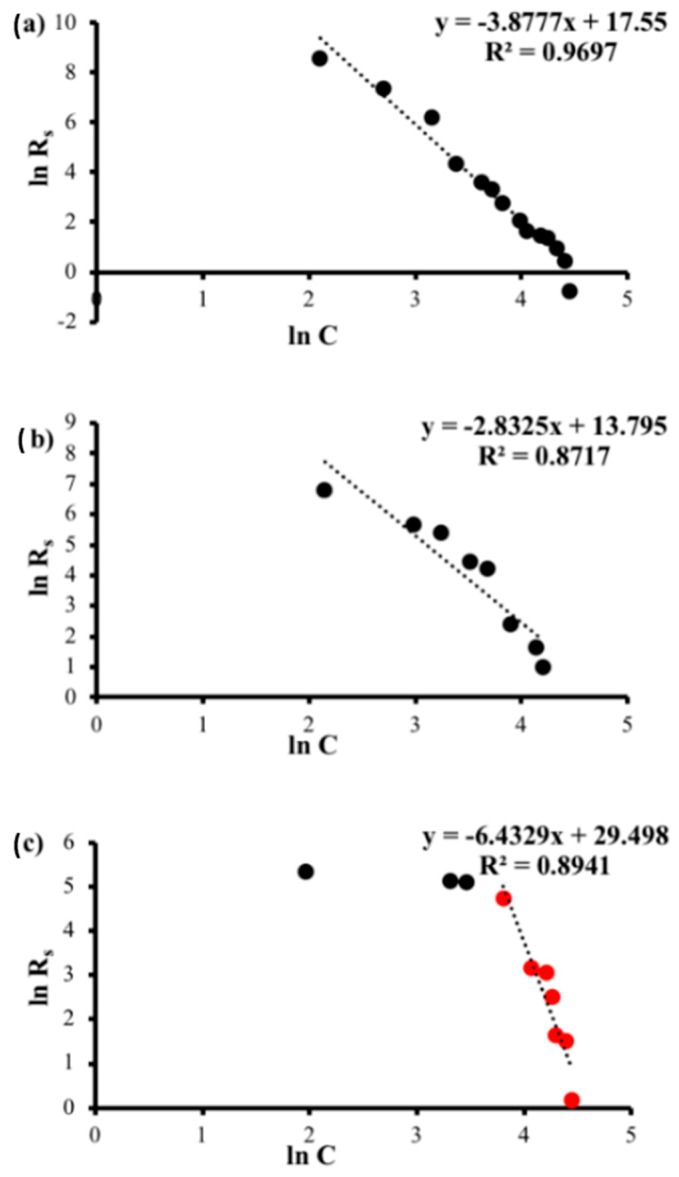
Natural logarithm of sheet resistance of sample I (**a**), sample II (**b**) and sample III (**c**) as a function of the natural logarithm of graphite concentration.

**Figure 11 polymers-14-04767-f011:**
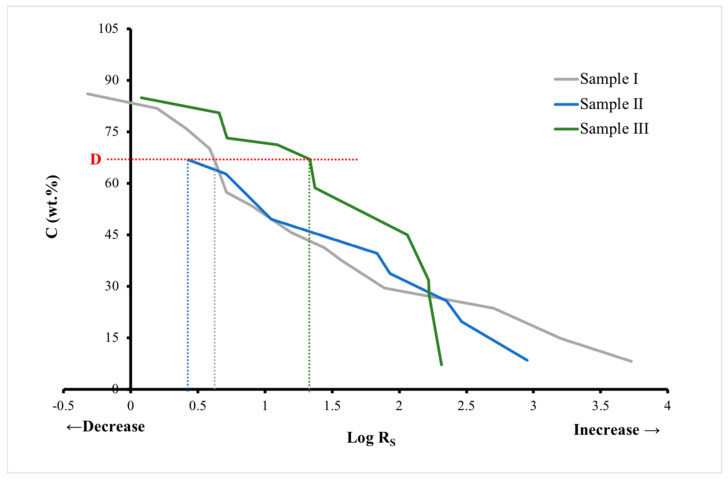
Investigation of efficiency of graphite cotton as electrical conductor. In point D, the conductive cotton treated with a mixture of graphite and DMSO (sample II) is more efficient than the other two samples because it achieves low resistances with significantly less graphite.

**Figure 12 polymers-14-04767-f012:**
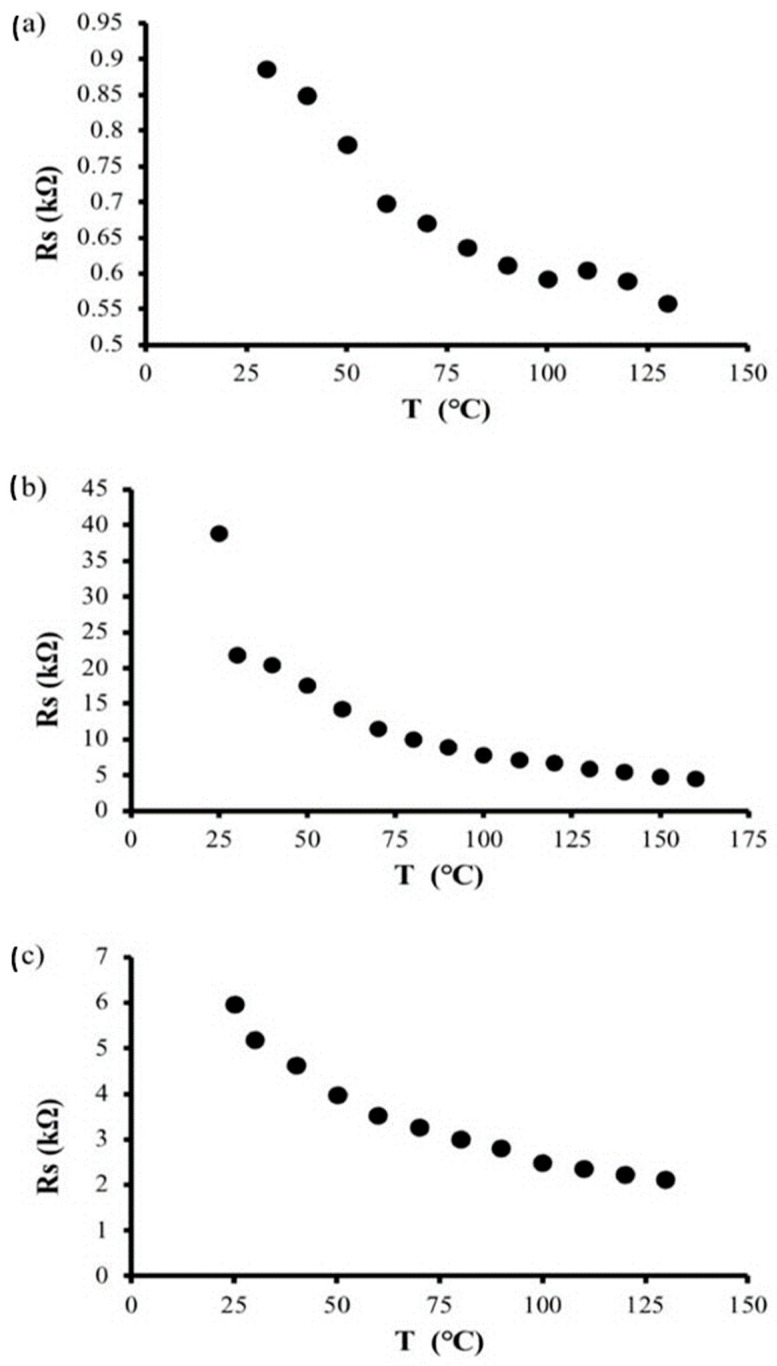
The sheet resistance of (**a**–**c**) for sample I, II and III, respectively, as a function of temperature.

**Figure 13 polymers-14-04767-f013:**
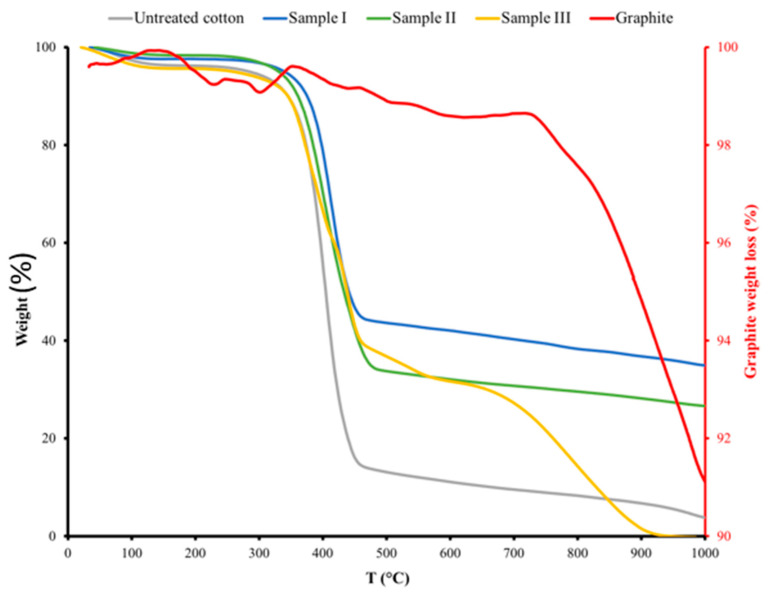
TGA analysis of graphite powder, untreated cotton, graphite cotton fabrics.

**Figure 14 polymers-14-04767-f014:**
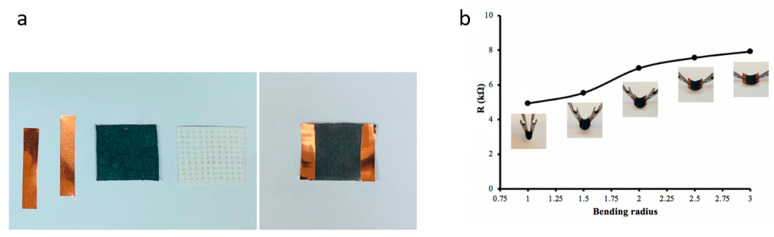
(**a**) graphite-cotton strain sensor fabrication (**b**) The resistance of cotton-based bending sensor during concave down bending.

**Figure 15 polymers-14-04767-f015:**
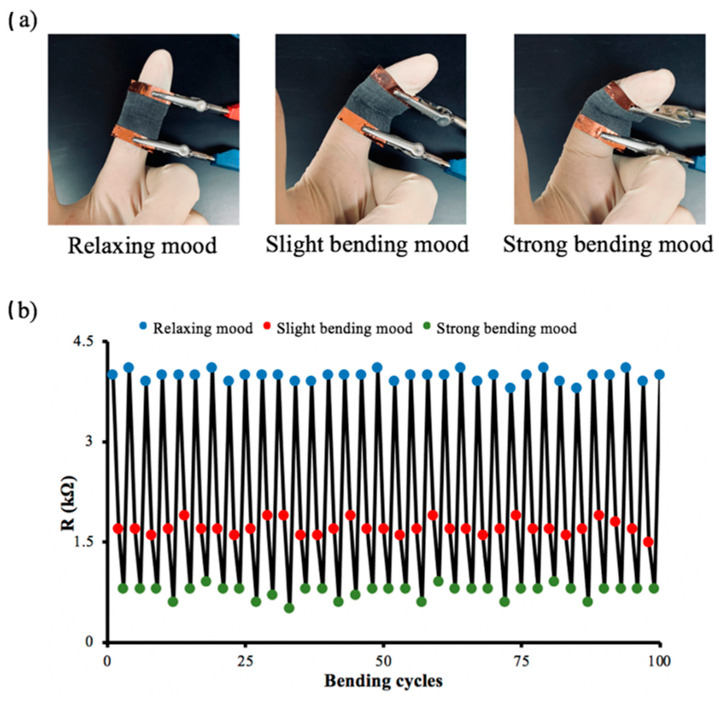
(**a**) The three different bending mood. (**b**) The resistance of sensor shows regular change in monitoring of the finger bends.

**Figure 16 polymers-14-04767-f016:**
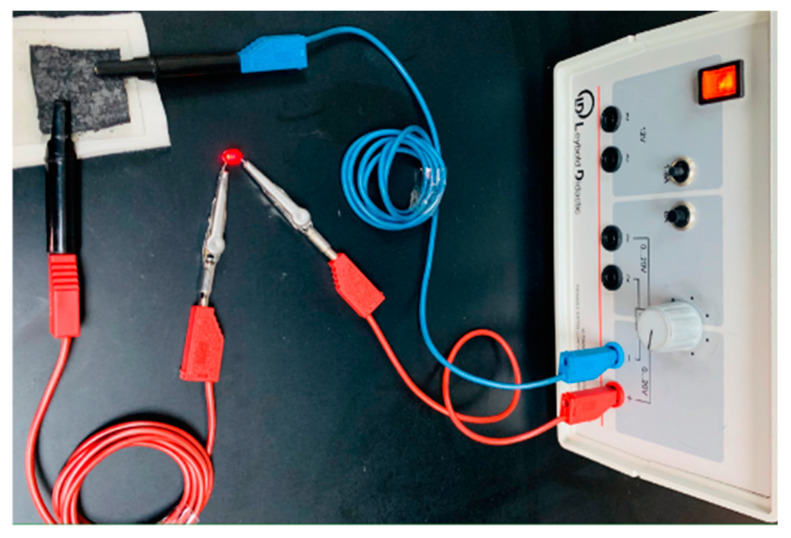
Graphite cotton is used as a conductive connector in an electrical circuit.

**Figure 17 polymers-14-04767-f017:**
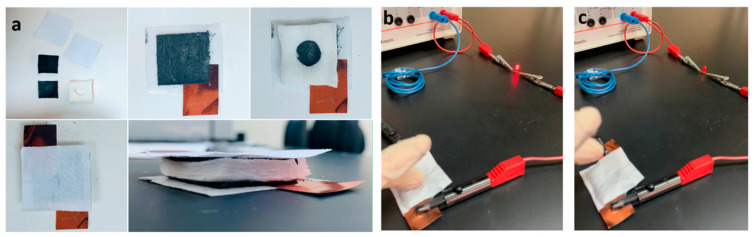
(**a**) graphite cotton fabric electrical switch fabrication (**b**,**c**) show an electrical circuit containing a graphite cotton fabric electrical switch in “on/off switch”. When pressure is applied, the light from LED turns on (**b**), while when pressure is removed, the light from LED turns off (**c**).

**Table 1 polymers-14-04767-t001:** Shows that the sheet resistance values of our work are lower than those reported in the literature for different fabrics infused with carbon-based materials, and how many bending cycles the fabrics used as strain sensors undergo.

Conductive Material Coated Fabric	Types of Substrate	Method	Electrical Connection	Strain Sensor (Bending Cycling)	Ref.
Graphite/graphene	PET fabric	interface trapping	3.6 kΩ/□	-	[44]
graphene oxide	fabric	Dipping-reduction	0.19 MΩ/□	-	[53]
graphene/waterborne/polyurethane	fabric	dip-coating	1.5 kΩ/□	-	[54]
polyacrylonitrile/graphite	fabric	doctor blade	4 kΩ	-	[55]
graphene oxide	cotton	sol-gel	6.70 MΩ/□	-	[56]
graphene oxide	polyester nonwoven	simple dip coating	330 Ω/□	50	[57]
Reduced graphene oxide	cotton	simple pad-dry	361.82 kΩ/□	within 160	[58]
PEDOT:PSS and graphene nanoflake	cotton	simple spray coating	~25 Ω/□	1000	[59]
graphite	Cotton	brush-coating-drying	1.197 kΩ/□	100	This study

**Table 2 polymers-14-04767-t002:** XRD calculations for untreated cotton, graphite and sample I, II and III at low, medium and high concentrations.

Sample	JCPD Card File No	$\mathrm{Diffraction}\;\mathrm{Angles},\;2\theta^{o}$	Miller Indices			Interplanar Spacing, d (nm)
			h	k	l	
untreated Cotton	00-050-2241	14.9879	1	−1	0	5.9061
		16.4866	1	1	0	5.3724
		22.7814	2	0	0	3.9002
Graphite	00-008-0415	26.49	0	0	2	0.3370
		44.48	1	0	1	0.2036
		54.54	0	0	4	0.1682
Low Concentration (Sample I)		14.9879	1	−1	0	5.9061
		16.4866	1	1	0	5.3724
		22.7814	2	0	0	3.9002
Medium Concentration (Sample I)		14.9879	1	−1	0	5.9061
		16.4866	1	1	0	5.3724
		22.7814	2	0	0	3.9002
		26.49	0	0	2	0.3370
High Concentration (Sample I)		22.7814	2	0	0	3.9002
		26.49	0	0	2	0.3370
Low Concentration (Sample II)		14.9879	1	−1	0	5.9061
		16.4866	1	1	0	5.3724
		22.7814	2	0	0	3.9002
		26.49	0	0	2	0.3370
Medium Concentration (Sample II)		26.49	0	0	2	0.3370
High Concentration (Sample II)		26.49	0	0	2	0.3370
Low Concentration (Sample III)		14.9879	1	−1	0	5.9061
		16.4866	1	1	0	5.3724
		22.7814	2	0	0	3.9002
		26.49	0	0	2	0.3370
Medium Concentration (Sample III)		26.49	0	0	2	0.3370
High Concentration (Sample III)		26.49	0	0	2	0.3370

**Table 3 polymers-14-04767-t003:** TGA analysis of graphite, untreated cotton and graphite cotton fabrics.

TGA Analysis	Untreated Cotton	Sample I	Sample II	Sample III		Graphite
				First Peak	Second Peak	
∆Y (%)	56.35	38.50	37.05	31.42	53.26	8.34
Onset (°C)	313.52	330.29	320.09	333.95	420.90	799.34
Endset (°C)	469.56	464.35	487.97	412.50	537.35	999.06
T_o_ (°C)	409.50	421.18	410.76	395.64	443.19	983.18

**Table 4 polymers-14-04767-t004:** Stability range of graphite, untreated and treated cotton fabric samples.

Samples	Stability Range
Graphite powder	100–800 °C
untreated cotton	100–340 °C
Sample I	100–350 °C
Sample II	100–343 °C
Sample III	100–340 °C

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
