# Peer review of "Construction of an Electrical Conductor, Strain Sensor, Electrical Connection and Cycle Switch Using Conductive Graphite Cotton Fabrics"

_polymers, 2022, doi:10.3390/polym14214767_

Round 1
Reviewer 1 Report
Journal: Polymers
Title: Construction of an Electrical Conductor, Strain Sensor, Electrical Connection and Cycle Switch using Conductive Graphite Cotton Fabrics
This article deals with a study on the construction of an electrical conductor, strain sensor, electrical connection, and cycle switch using conductive graphite cotton fabrics. The authors successfully prepared conductive textiles by applying a graphite dispersion to cotton fabric using a simple brush-coating-drying method and the solvents of dimethyl sulfoxide, dimethyl formamide, and a solvent mixture of both. Moreover, The report shows that these graphite cotton fabrics can be used as electrical interconnects in electrical circuits without any visible degradation of the conductive cotton.
By reading the overall manuscript, I suggest revision the following queries,
(1)What does the conductive, strain, electrical connection, or cycle switch properties of sensors compared with other published materials or commercial materials? or the authors can consider adding the figure/table compared with other references?
(2) How about the stability of the bending cycle test over 100 times?
(3)Authors can consider citing the following reference, which can strengthen the present manuscript.
Nano Energy, 2022, 101, 107592.
https://doi.org/10.1016/j.nanoen.2022.107592
Reactive and Functional Polymers, 2022, 181, 105421.
https://doi.org/10.1016/j.reactfunctpolym.2022.105421
Nanomaterials 2022, 12(12), 2039.
https://doi.org/10.3390/nano12122039

Author Response
please check the attach file

Reviewer 2 Report
In this work, the authors presented the preparation of carbon-based conductive textiles for the design and fabrication of flexible strain sensor. A simple method was developed to fabricate graphite-coated cotton fabrics by applying conductive graphite material to the cotton fabric with a brush. The fabricated sensor exhibited good electrical conductivity and excellent reproducibility after numerous bending cycles, providing a favorable basis for application in wearables. In addition, this manuscript represents a direct continuation of the author’s research in the field of conductive cotton fabrics. I find the manuscript to be well structured and written in fluent English. I recommend that the submitted manuscript be accepted for publication after making minor changes as listed below:
· Authors are advised to review the entire manuscript in detail and to correct minor grammatical errors.
· Please provide further discussion or citation as to why DMSO and DMF were chosen for graphite dispersion. Since DMF is flammable and hazardous to health solvent, there is question about how it is suitable during the fabrics drying process? Please comment on this regarding to lines145-148.
· Is „brush-coating-drying“ method of manufacturing sensor suitable for mass production?
· Lines 240-242, please provide a supporting citation for the assigned vibration modes. To the revewers knowledge, there is no triple covalent bond between the carbon atoms in the graphite structure.
· Figure 12: In the graphical represention of the measured sheet resistance for sample I, why is the temperature expressed in Kelvin, while the other two plots are in degrees Celisius?
Author Response
please check the attach file
